# A study of the effect of aerosols on surface ozone through meteorology feedbacks over China

Yawei Qu[1,2,3], Apostolos Voulgarakis[2,4], Tijian Wang[1], Matthew Kasoar[2], Chris Wells[2], Cheng Yuan[5], Sunil Varma[2], Laura Mansfield[2]

[1]School of Atmospheric Sciences, Nanjing University, Nanjing, China
[2]Leverhulme Centre for Wildfires, Environment and Society, Department of Physics, Imperial College London, London, UK
[3]School of Intelligent Science and Control Engineering, Jinling Institute of Technology, Nanjing, China
[4]School of Environmental Engineering, Technical University of Crete, Crete, Greece
[5]School of Atmospheric Physics, Nanjing University of Information Science & Technology, Nanjing, China

*Correspondence to*: Apostolos Voulgarakis (a.voulgarakis@imperial.ac.uk) and Tijian Wang (tjwang@nju.edu.cn)

**Abstract.** Interactions between aerosols and gases in the atmosphere have been the focus of an increasing number of studies in recent years. Here, we focus on aerosol effects on tropospheric ozone that involve meteorological feedbacks induced by aerosol-radiation interactions. Specifically, we study the effects that involve aerosol influences on the transport of gaseous pollutants and on atmospheric moisture, both of which can impact ozone chemistry. For this purpose, we use the UK Earth

System Model (UKESM1) with which we performed sensitivity simulations including and excluding the aerosol direct radiative effect (ADE) on atmospheric chemistry, and focused our analysis on an area with high aerosol presence, namely China. By comparing the simulations, we found that ADE reduced the shortwave radiation by 11% in China, and consequently led to lower turbulent kinetic energy, weaker horizontal winds and a shallower boundary layer (with a maximum of 102.28 m reduction in north China). On the one hand, the suppressed boundary layer limited the export and diffusion of pollutants, and

increased the concentration of CO, $SO_2$, NO, $NO_2$, $PM_{2.5}$ and $PM_{10}$ in the aerosol rich regions. The $NO/NO_2$ ratio generally increased and led to more ozone depletion. On the other hand, the boundary layer top acted as a barrier that trapped moisture at lower altitudes and reduced the moisture at higher altitudes (the specific humidity was reduced by 1.69% at 1493 m averaged in China). Due to reduced water vapor, fewer clouds were formed, and more sunlight reached the surface, so the photolytical production of ozone increased. Under the combined effect of the two meteorology feedback methods, the annual average ozone

concentration in China declined by 2.01 ppb (6.2%), which was found to bring the model in closer agreement with surface ozone measurements from different parts of China.

## 1 Introduction

Atmospheric aerosols could change the Earth's radiation budget by scattering and absorbing the incoming solar radiation, which is known as the aerosol direct radiative effect (ADE; Myhre et al., 2013). Scattering aerosols, such as sulfate, nitrate,

organic carbon and sea-salt, reflect the shortwave radiation and lead to the negative radiative forcing (Choi and Chung, 2014; Hollaway et al., 2019); while absorbing aerosols, such as black carbon (BC) and dust, absorb sunlight and lead to the positive

radiative forcing at the top of the atmosphere. Absorbing aerosols heat the atmosphere but cool the Earth's surface by reducing the downward solar radiation. Aerosols can also influence the radiation by aerosol-cloud interactions, i.e. the aerosol indirect effect (AIE). By acting as the condensation and nucleation sites, aerosols are related to clouds microphysical development.

When there are more aerosols, there will be more clouds but smaller droplets, leading to brighter clouds and more shortwave radiation being reflected back to the space (Twomey, 1974). In addition, the higher in terms of number but smaller cloud droplets mean delayed precipitation and a longer lifetime of clouds (Albrecht, 1989; Stevens and Feingold, 2009).

The direct radiative effect of aerosols plays an important role in ozone chemistry. Tropospheric ozone is produced mainly by the photolysis of $NO_2$ ($NO_2$ + hv -> NO + $O^3P$, followed by $O^3P$ + $O_2$ -> $O_3$), and can also be destroyed by photolysis ($O_3$ +

hv -> $O_2$ + $O^1D$). The photodissociation reaction rate (photolysis rate) is highly related to the shortwave radiation, which could be influenced by aerosols (He and Carmichael, 1999). Due to ADE, the photolysis rates of $NO_2$ ($J_{NO2}$) and $O^1D$ ($J_{O1D}$) have been found to be reduced by 3% to 30% in Europe (Real and Sartelet, 2011), Texas (Flynn et al., 2010), Mexico (Castro et al., 2001; Li et al., 2011b), Russia (Péré et al., 2015) and China (Hollaway et al., 2019; Wang et al., 2019; Xu et al., 2012), with consequent effects on ozone concentration.

In addition to its impact on photochemical reactions, ADE can affect meteorological conditions by influencing regional energy balance and the vertical structure of the planetary boundary layer (PBL). The PBL is the bottom layer of the atmosphere that connects it to the Earth's surface (Stull, 1988). The air pollutants in the troposphere, including ozone and its precursors, are primarily distributed in the PBL and can be redistributed by turbulent mixing, advective (horizontal) transport, and vertical diffusion (Li et al., 2018). The top of the PBL also acts as a barrier, which prevents aerosols, water vapour and other chemicals

to be exchanged between the PBL and the free troposphere. The radiative effect of aerosols reduces downward solar radiation and therefore cools the Earth's surface, which leads to lower turbulent kinetic energy and lower PBL height (Li et al., 2017b; Wilcox et al., 2016). A high aerosol loading has also been found to be responsible for a delayed PBL formation in the morning and an earlier PBL collapse in the afternoon (Barbaro et al., 2014). Meanwhile, a more stable boundary layer could slow down the atmospheric movement and make it less likely for pollutants to be transported and dispersed. The relationship between

PBL characteristics and pollution events has been highlighted for various regions around the world, e.g. Spain (Adame et al., 2015), Paris (Dupont et al., 2016), India (Nair et al., 2018; Patil et al., 2014), and China (Gao et al., 2015; Liu et al., 2020; Miao and Liu, 2019; Qu et al., 2017). Though boundary layer ozone is less restricted to PBL due to its relatively long lifetime (Hayashida et al., 2018; Verstraeten et al., 2015), the consumers and precursors of ozone could be influenced by this meteorological feedback between aerosols and PBL (Nguyen et al., 2019), therefore influencing ozone itself.

Another possible mechanism that is even less direct is the following: By weakening atmospheric movement and lowering the PBL, water vapor increases in the PBL and becomes difficult to be transported from the PBL to the free troposphere (Hansen et al., 1997). The reduced humidity will limit the development of clouds, thus allowing more sunlight to reach the surface (Wilcox et al., 2016). The photolysis rates that drive atmospheric photochemical reactions thereby vary and result in the changes in air pollutants and ozone levels (Johnson, 2003). Tang et al. (2003) found that clouds have large impact on

tropospheric photolysis rates and ozone concentration, which lead to a decrease of $J_{NO2}$ by 20% and ozone by 1.2% below

clouds along the TRACE-P flight paths. For global scale, Liu et al. (2006) found that clouds have smaller impact on photolysis rates (less than -5%). Using the Cambridge p-TOMCAT chemical transport model (CTM), Voulgarakis et al. (2009a, 2009b) showed that clouds have modest effect on global average ozone, but have a larger impact in the areas with higher cloud cover. Apart from the radiative effect, aerosol can also influence ozone through the chemical effect, which is the heterogeneous

reaction. By reacting with ozone, nitrogen oxides, OH, $HO_2$, $H_2O_2$, etc., mineral and carbonaceous aerosols can affect ozone concentration directly and indirectly (Bauer, 2004; Ramachandran, 2015; Tang et al., 2017).

Based on the above, Figure 1 summarises five possible influences of aerosols on ozone: 1) heterogeneous reactions, 2) directly changing photolysis rate (ADE-PHO), 3) influencing the distribution of atmospheric pollutants, including ozone and its precursors through meteorological feedbacks (ADE-POL), 4) changing the photolysis rates through influencing moisture

transport (ADE-MOI), 5) modifying clouds and, consequently, chemistry via microphysics, i.e. the aerosol indirect effect (AIE). ADE-POL and ADE-MOI can be thought as the meteorological mechanisms that are both dominated by atmospheric transport feedbacks. Regarding the chemical effect, the impact of heterogeneous reactions on ozone has been investigated through a lot of laboratory and model studies (Bauer, 2004; Griffiths and Anthony Cox, 2009; Stewart and Cox, 2004; Tang et al., 2017). Regarding radiative effects, though the aerosol radiative influence on climate has been widely studied, the less

abundant studies of their influence on ozone mainly focus on the ADE-PHO (Li et al., 2011a; Qu et al., 2019) and AIE (Hall et al., 2018; Voulgarakis et al., 2009a; Wild et al., 2000), while ADE-POL and ADE-MOI are much less discussed in the literature. Therefore, in this paper, we exclude the impact of heterogeneous reactions, direct photochemical or microphysical effects, and focus on the combined effect of ADE-POL and ADE-MOI, i.e. the meteorological feedback, on tropospheric ozone. This enables a better understanding of the interaction between aerosols and ozone in China, and provides a more comprehensive

scientific background for the control of atmospheric particulate matter, ozone and photochemical pollution.

A set of sensitivity simulations has been performed, by using the first version of the UK Earth System Model UKESM1, to investigate the influence of meteorological feedbacks of aerosols on ozone in different parts of East Asia. Section 2 introduces the observational data and numerical model set-up that is used in this study. Section 3.1 evaluates the performance of the numerical model by comparing to observational data. Section 3.2 discusses the aerosol-PBL feedback. Section 3.3

demonstrates the impact of ADE on atmospheric pollutants (ADE-POL). Section 3.4 demonstrates the impact of ADE on moisture, clouds and then photolysis rates (ADE-MOI). Section 3.5 discusses the combined effect of ADE-POL and ADE-MOI on ozone. The conclusions and perspective are presented in Section 4.

## 2 Methods

### 2.1 Observation

Air pollutant concentrations at more than 1000 national ambient air quality monitoring sites are released by the Ministry of Environmental Protection (MEP) of China and can be downloaded from the China National Environmental Monitoring Center (CNEMC, http://www.cnemc.cn/sssj/). The technical requirements for the monitoring system including the composition,

installation, operation, maintenance and data quality control are addressed in the China Environmental Protection Standards "HJ 193-2013" (http://www.mee.gov.cn/ywgz/fgbz/bz/bzwb/jcffbz/201308/W020130802493970989627.pdf) and "HJ 655-

2013" (http://www.mee.gov.cn/ywgz/fgbz/bz/bzwb/jcffbz/201308/W020130802492823718666.pdf). In this paper, the hourly concentrations of $SO_2$, $NO_2$, $O_3$, CO, $PM_{2.5}$ and $PM_{10}$ at 1412 monitoring sites in 2014 were utilised from CNEMC. The locations of the monitoring sites are shown in Fig. 2. The observational data was used to evaluate the simulated air pollution over China.

## 2.2 UKESM1-AMIP

The 1[st] version of the United Kingdom Earth System Model (UKSEM1) is jointly developed by Natural Environment Research Council (NERC) and the Met Office Hadley Centre and has been released in Feb 2019 (Sellar et al., 2019). UKESM1 is based on the physical climate model HadGEM3 (Hewitt et al., 2011; Kuhlbrodt et al., 2018; Williams et al., 2018) and couples additional components, including the land biogeochemistry model (JULES; Clark et al., 2011), the UK Chemistry and Aerosols model (UKCA; Archibald et al., 2020; Mulcahy et al., 2018), the dynamic vegetation model (TRIFFID; Cox, 2001), and the

Interactive ocean biogeochemistry model (MEDUSA; Yool et al., 2013). In this study, we used its atmosphere-only (UKESM1-AMIP) version to study the radiative effect of aerosols on ozone. Unlike the fully coupled UKESM1, the atmosphere-only configuration doesn't include ocean and sea ice models (NEMO/CICE), MEDUSA and TRIFFID. Instead, UKESM1-AMIP uses prescribed, observation-based sea surface temperatures and sea ice data (https://pcmdi.llnl.gov/mips/amip/). The model input for vegetation and surface ocean biology fields are provided by the UKESM1 CMIP6.

The core atmospheric model of UKESM1-AMIP is the 11.1 version of the Met Office Unified Model (UM; Walters et al., 2019), in which the atmospheric chemistry and aerosols are modeled by UKCA. The new Global Model of Aerosol Processes (GLOMAP-mode; Mann et al., 2010) is a size-resolved aerosol microphysics model. It is used for aerosol simulation in UKCA, including the mass and number of sulfate, black carbon, organic carbon, and sea salt. Dust aerosols are not available yet in GLOMAP-mode and so a bin scheme for mineral dust (Woodward, 2001) is used. The photolysis scheme in UKCA is Fast-

JX (Telford et al., 2013), which provides the full scattering calculation for 18 wavelength bins over 177-850 nm. Fast-JX allows the calculation of the interactive photolysis rates in the troposphere (Wild et al., 2000) and improves the calculation of photolysis rates in stratosphere (Bian and Prather, 2002). In order to focus on ADE-POL and ADE-MOI effects (see Sect. 1), Fast-JX has not been coupled with the GLOMAP-mode aerosol scheme, which means that photolysis rates are independent of the aerosol loading (Sellar et al., 2019).

Two sensitivity simulations were performed to evaluate the radiative effects of aerosols on ozone: 1) $EXP_{radon}$: including the aerosol direct radiative feedback on atmospheric chemistry, and 2) $EXP_{radoff}$: without this radiative feedback. The simulation covers one year, i.e. from the 1[st] Jan 2014 to the 31[st] Dec 2014. The atmospheric horizontal resolution of UKESM1-AMIP is N96 (~140 km) and the vertical resolution is 85 levels. Emissions are the year 2014 CMIP6 emissions for all runs.

# 3 Results

## 3.1 Model evaluation

The model performance was evaluated by comparing the simulation results at the surface layer with the ground-based observations. The simulation with radiation feedback ($EXP_{radon}$) was carried out as the control test. Figure 3 shows annual average concentrations of $O_3$, CO, $NO_2$, $SO_2$, $PM_{2.5}$ and $PM_{10}$ simulated in $EXP_{radon}$ along with the concentrations observed at monitoring sites. Pearson's correlation coefficient (R) and mean bias error (MB) are shown in Table 1, using daily average concentration data. In terms of the spatial distribution, the simulation results are found to be in fairly good agreement with the observations. With the economic and industrial development in north and east China, anthropogenic emissions lead to increased air pollution in these areas (Li et al., 2017a). The model captures the high $SO_2$, CO, and $NO_2$ concentrations and the high aerosol loading in north and east China. However, the model produces much higher $SO_2$ concentrations than the observations, most likely due to an overestimation of the emissions. Under the clean air policies, the $SO_2$ emission has declined by 62% during 2010–2017 (Zheng et al., 2018), but the CMIP6 emissions do not capture this reduction, with 2014 $SO_2$ emissions being higher by 48% when compared to the region-specific Multi-resolution Emission Inventory for China (MEIC) (Paulot et al., 2018). For the spatial distribution of ozone, the model is in good agreement with observations. The simulated ozone concentration is well correlated with the observed values, with R reaching a maximum of 0.8 in the JJJ area. The radiation effect improved the model performance in China. When including the meteorological feedback of radiation effect, the average MB of ozone dropped from 10.03 to 5.63, while the average R remained the same (Table 1). In most areas, the correlation between observed and simulated CO, NO, $SO_2$, and particulate matter were higher in $EXP_{radon}$ than in $EXP_{radoff}$, indicating that including these effects improves the simulation of tropospheric pollutants. Subsequently, we examine these effects in more detail.

## 3.2 Aerosol effect on meteorology

The aerosol effect on meteorology was assessed by taking averages over the 1-year simulation and taking the difference between $EXP_{radon}$ and $EXP_{radoff}$. Figure 4 shows the changes in net downwelling surface shortwave radiation, turbulent kinetic energy, planetary boundary layer height (PBLH), and 10-m wind due to the direct effect of aerosols on radiation. Shortwave radiation is generally reduced due to aerosols over China and the largest reduction is found in more aerosol-rich parts of the country (Fig. 3l,m), i.e. north and east China. Shortwave radiation was reduced by 30.24 W m$^{-2}$ (18.85%), 19.73 W m$^{-2}$ (12.98%), 20.45 W m$^{-2}$ (11.22%) and 16.27 W m$^{-2}$ (13.53%) in JJJ, YRD, PRD and SCB, respectively (Figure 4a). The high-resolution regionally-focused WRF-Chem simulation performed by Wang et al. (2016) similarly showed that due to ADE, the solar radiation in China decreased by 20 W m$^{-2}$, and the percentage changes ranged from 11.7% to 14.3% in different areas. A decreased downwelling solar radiation could cool the surface and cause weaker thermal turbulence in the boundary layer (Liu et al., 2018; Quan et al., 2013). The temperature at 1.5 m is found to be reduced in the north China plain and southwest China (Fig. S1a) due to the radiation changes. TKE (Fig. 4b) showed the largest change in north China (JJJ), with a decline of

0.12 m$^2$ s$^{-2}$ (33.43%), which is consistent with China's largest SW radiation change area. This is in line with the findings of Wang et al., (2020), who found that during a haze episode in winter, the TKE in Beijing declined by 0.1–0.7 m$^2$ s$^{-2}$ due to the aerosol-included effect. The reduction of TKE in YRD reaches 23.09% in our findings, which is the second-highest TKE reduction region in China.

The growth of boundary layer mainly depends on the atmospheric thermal structure and turbulent exchange intensity (Garratt, 1994; Serafin et al., 2018). Owing to the reduced solar radiation and TKE, the development of the PBL was supressed, and resulted in a shallower and more stable boundary layer (Fig. 4c,d). In north China (JJJ), the annual average planet boundary layer height (PBLH) was reduced by 102.28 m (22.01%) due to the ADE. Observations in this area also showed that the average PBLH was reduced by 334-710 m during severe pollution periods compared to clean days (Tang et al., 2016; Zhang

et al., 2015). The annual delination of PBLH in YRD was 53.39 m (16.26%), and this reduction was consistant with the WRF-Chem simulation by Wang et al. (2016), who found that PBLH in East china decreased by 75.2 m in spring, while in other regions of China it decreased by 75-138m. Using the WRF-CMAQ model, Nguyen et al. (2019) also found that the ADE could reduce the annual average PBLH in East Asia by 46.47 m (8.13%). The lower boundary layer caused by aerosols is usually also accompanied by calm winds and higher relative humidity values (Yin et al., 2019). Here, the 10-m wind is found to be

lowered by 1% to 7.5% (Fig. 4d), and relative humidity at 1.5 m increased with a maximum of 5.7% (Fig. S1b). The variations in wind and boundary layer stability would influence horizontal transport and pollutants and moisture accumulation, as well as the vertical dispersion and the exchange of clean air with the free troposphere.

**3.3 Impact of meteorology feedback via atmospheric pollutants (ADE-POL)**

The aerosols direct radiative feedback was found to reduce solar radiation which resulted in the suppression of PBL height and
turbulent intensity, while the suppressed PBL in turn limits the export and diffusion of pollutants. Figure 5 shows the influence of ADE on surface CO, SO$_2$, NO, NO$_2$, PM$_{2.5}$ and PM$_{10}$ concentrations. Overall, pollutant concentrations increased when including aerosols, due to the decreasing wind speeds and PBLH. The CO increase caused by ADE averaged over China was 11.04%, with the biggest changes appearing in north China (JJJ), east China (YRD), and central China (up to 12.25-16.17%). The distribution of SO$_2$, NO and NO$_2$ changes is similar to that of CO, with increases of 5.66-38.99%, 7.71-55%, and 2.78-

40.63%, respectively. For fine and coarse aerosols (PM$_{2.5}$ and PM$_{10}$), the increases are between 9.5% and 18.6% in the four selected areas and the spatial distributions of changes are similar to those of gaseous pollutants. Changes in gas and aerosol pollutants were the result of the changes in meteorological conditions. The shallower PBLH reduced the vertical dispersion and compressed the pollutants in PBL, resulting in higher surface pollutant concentrations. The increased boundary layer stability and reduced wind speed also led to the accumulation of pollutants at their emission sources. The spatial distribution

of the changes in pollutant concentration is similar to the spatial distribution of meteorological conditions changes and emission sources. With a larger population and more developed industries, north and east China were considered to be the high-emission areas of the country (Wang, 2015; Zheng et al., 2018). These areas are more sensitive to the accumulation of pollutants and showed a stronger increase in the pollutants concentrations due to aerosol effects. Western China is less developed than the

eastern parts, and its population and anthropogenic emissions are also lower (Saikawa et al., 2017; Shi et al., 2014). As a result of that, the ADE in west China caused a small increase and even a decrease in pollutant concentrations. In southwest China, SCB is more developed than the surrounding cities, and its bowl-shaped topography helps trap air pollutants (Ning et al., 2018). More pronounced increases in pollutants' concentrations are also found in this area, but the magnitude is lower than that in JJJ and YRD. Changes in air pollutants (including NO and $NO_2$) in different regions affect the ratio of $NO/NO_2$, which is related to the loss and the production process of ozone. The change in $NO/NO_2$ and ozone will be further discussed in section 3.5.

## 3.4 Impact of meteorology feedback via moisture (ADE-MOI)

The changes in boundary layer stability and PBLH would not only contribute to the pollutant accumulation, but also linked to the moisture accumulation. The change in horizontal water vapor flux over the land area is small (Fig. S2). However, a low PBLH could limit the vertical transport of water vapor from the boundary layer to the free troposphere. Figure 6 shows the vertical profile of changes of specific humidity in different parts of China. In most seasons, climatological specific humidity increases in the lower troposphere and drops in the higher layers. In JJJ, the area most affected by ADE, the surface moisture content increases more when comparing $EXP_{radon}$ with $EXP_{radoff}$, with a maximum change of 4.28% ($6.55 \times 10^{-4}$ kg kg$^{-1}$) in June. The annual mean specific humidity decreased by a maximum of 1.69% ($1 \times 10^{-4}$ kg kg$^{-1}$) at 1493 m in China.

When more water vapor was trapped in the lower troposphere, there would be less moisture to form cloud in the upper layers (Allen et al., 2019). The annual average cloud amount decreases by 4% due to aerosol effects on radiation over the whole country (Fig. 7). The area with the largest decline is YRD with a percentage of 5%. The cloud optical depth also drops by 7%-15.6% in China, with the regional distribution of changes being similar to the cloud amount changes. Clouds attenuate solar radiation, leading to diminished photolysis rates beneath the cloud (Tang et al., 2003; Voulgarakis et al., 2009a, 2009b, 2010). Therefore, the increased water vapor in PBL results in higher photolysis rates by reducing clouds. However, the increased water vapor in PBL will also enhance extinction by aerosol hygroscopic growth, which results in lower photolysis rates. Figure 7 shows that surface photolysis rates $J_{NO2}$ and $J_{O1D}$ both increase, which means, comparing to the aerosol hygroscopic growth, the aforementioned cloud reductions is the dominant effect. The national average $J_{NO2}$ and $J_{O1D}$ rose by 4.1% and 3.3%, respectively. SCB is the region with the largest increase in $J_{NO2}$ and $J_{O1D}$, with percentage increases of 8% and 7.9%, respectively. The increase in $J_{NO2}$ and $J_{O1D}$ could lead to an increase/decrease in ozone concentration.

## 3.5 $O_3$ changes due to aerosol's meteorology feedback

The meteorological feedbacks that we study, ADE-POL and ADE-MOI, may have contrasting effects on ozone. For ADE-POL, the relationship between NO and $NO_2$ concentrations could be used to predict the changes in ozone concentration, because NO and $NO_2$ lead to the loss and production of ozone, respectively. Figure 8 shows the annual average $NO/NO_2$ ratio changes. In the aerosol polluted areas, i.e. north China, YRD, PRD, SCB and central China, the $NO/NO_2$ ratio increased with the highest value of 0.17. West China, south China (exclude PRD) and north-east China were less influenced by ADE-POL,

and the NO/NO$_2$ ratio showed a small change. The observations in Germany (Melkonyan and Kuttler, 2012), Brazil (De Souza et al., 2017) and China (Han et al., 2011) have demonstrated that an increasing NO/NO$_2$ ratio could consume more ozone and reduce ozone concentration. In ADE-MOI, $J_{NO2}$ and $J_{O1D}$ were both increased due to the cloud amount and optical depth changes. Tang et al., (2003) found that the $J_{NO2}$ was more sensitive to cloud than $J_{O1D}$ and most other photolysis rates, and the decrease of cloud cover could lead to higher net ozone production below the cloud layer. Therefore, changes in the atmospheric
water content and subsequent cloud changes could lead to local increases in surface ozone concentration.

These two opposite effects compete against each other, resulting in different ozone changes in different regions and seasons. Figure 9 presents the seasonal changes of NO/NO$_2$ ratio (representing the ADE-POL effect), photolysis rates (representing the ADE-MOI effect) and ozone concentration in the four selected regions and in the whole country. The increase in NO/NO$_2$ ratio dominates the ozone changes and diminished the surface ozone concentration in all seasons and regions, except for
February in the YRD and SCB regions, when the ADE-MOI effect overwhelmed the ADE-POL effect. The magnitude of ozone percentage change appears to depend on the relative magnitude of the NO/NO$_2$ ratio changes and the photolysis rates change. In northern cities, such as JJJ, the monthly variation in ozone changes showed a double-peak pattern, with the largest decline in spring and autumn, while in south China, the change in ozone only reaches its largest reduction in winter. The latitudes of YRD and SCB are in between the latitudes of JJJ or PRD, and therefore the seasonal patterns are not as clear as
for JJJ or PRD. In YRD, the combined effect leads to ozone changes ranging from -5 ppb to 0.07 ppb. Xing et al., (2017) found that the meteorology changes reduced the surface concentration of ozone in east China in January by 5-24 μg m$^{-3}$ (2.33-11.19 ppb). The reason for the difference might be that they did not include the positive feedback of ADE-MOI when analyzing meteorological effects. The reaction flux changes in Fig. S3 shows that, on annual average, the combined effect of ADE-POL and ADE-MOI led to more ozone consumption than ozone production, suggesting that ADE-POL dominates. Figure 10 shows
the spatial distribution of annual average ozone changes. The region with the highest ozone reduction is consistent with the region of largest NO/NO$_2$ ratio increase. Ozone concentration was found to decrease by 3.84 ppb (14.9%), 2.45 ppb (8.7%), 1.48 ppb (4.3%), and 1.78 ppb (7.1%) in JJJ, YRD, PRD and SCB on annual average, and it decreased by around 2.01 ppb (6.2%) averaged over the whole country.

**4 Conclusions**

In this paper, we used a coupled global earth system model, UKESM1-AMIP, to evaluate the influence of aerosol's meteorology feedback on tropospheric ozone over China. Aerosols reduced surface net downward shortwave radiation by 11% through the scattering and absorbing effect, and reduced the surface turbulent kinetic energy by 16.7%. The boundary layer was therefore less heated and developed, the height of which was found to decrease by 102.28 m in north China. The meteorology changes in the lower troposphere can influence the dispersion and mixing of pollutants (ADE-POLL effect) and
moisture (ADE-MOI effect). Gaseous pollutants such as CO, SO$_2$, NO, NO$_2$ all increased in the aerosol rich regions, and particulate matter (PM$_{2.5}$ and PM$_{10}$) increased by 9.5%-18.6% in the four selected areas. Different changes in NO and NO$_2$

affect the NO/NO$_2$ ratio, which is related to the loss and the production process of ozone. Moisture was found to be more trapped in the boundary layer, with specific humidity increasing in the PBL, and the strongest effects found in June in JJJ (4.28%). With more moisture accumulated near the ground, less moisture was transported to higher layers to form clouds. The

cloud amount reduced by 4% and clouds became more transparent. The photolysis rates for NO$_2$ and O$^1$D were thereby found to be increased by 4.1% and 3.3%, respectively.

Increased NO/NO$_2$ ratio (ADE-POL) consumes more ozone, while increased photolysis rate (ADE-MOI) produces more ozone. The net magnitude of ozone change due to aerosols is linked to the relative magnitude of the NO/NO$_2$ ratio change and the photolysis rates change. In general, the NO/NO$_2$ change dominated the ozone concentration change and led to reduced annual

average ozone in China, around 2.01 ppb (6.2%).

Overall, our study reveals that, except for the direct effect through photolysis rates changes, ADE can influence ozone concentration through two meteorological mechanisms: one is to affect the abundances of atmospheric pollutants, including ozone consumers and producers (ADE-POL); and the other is to affect the vertical transmission of water vapour, thus affecting the optical characteristics of clouds, and therefore ozone photochemical production through photolysis (ADE-MOI). The

combined effect and relative importance of meteorological feedbacks, direct photolysis influences, and microphysical influences needs to be assessed in a future study.

## Data availability

The data used in this study are available upon request from Yawei Qu (yawei_qu531@163.com).

## Author contributions

YQ designed the research study, ran model simulations and performed the data analysis under the close supervision of AV, with some additional supervisory support from TW. TW provided access to CNEMC data. MK, CW, SV and LM offered continued guidance and technical support on UKESM simulation. YQ wrote the original manuscript and AV, TW and CY reviewed the manuscript.

## Competing interests

The authors declare that they have no conflict of interest.

## Acknowledgement

This work is supported by the National Natural Science Foundation of China (41621005, 42077192), National Key Basic Research Development Program of China (2016YFC0203303, 2020YFA0607802). Yawei Qu is supported by the China

Scholarship Council (201806190151). Apostolos Voulgarakis is partially funded by the Leverhulme Centre for Wildfires,
Environment and Society through the Leverhulme Trust, grant number RC-2018-023. UKESM1-AMIP model simulations were performed using the MONSooN (Met Office and NERC Supercomputing Nodes), a shared HPC platform within a collaborative computing environment, which is developed by Met Office and NERC. Also, we wish to thank Luke Abraham from University of Cambridge and Mohit Dalvi from the UK Met Office for their support with using the UKESM model.

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

**Table 1: Statistical matrix for simulated and observed pollutant concentrations. (a) The correlation coefficient (R) and mean bias (MB) between observation and simulation in EXP$_{radoff}$. (b) The temporal correlation coefficient (R) and mean bias (MB) between observation and simulation in EXP$_{radon}$. The statistics are based on the daily average concentrations in 2014.**

(a)

| | $O_3$ | | CO | | $NO_2$ | | $SO_2$ | | $PM_{2.5}$ | | $PM_{10}$ | |
|---|---|---|---|---|---|---|---|---|---|---|---|---|
| | R | MB | R | MB | R | MB | R | MB | R | MB | R | MB |
| JJJ | 0.83 | -5.63 | 0.61 | 0.11 | 0.27 | 10.49 | 0.47 | 105.5 | 0.44 | -3.06 | 0.33 | -49.37 |
| YRD | 0.55 | -8.13 | 0.49 | 0.12 | 0.29 | 2.67 | 0.39 | 56.59 | 0.18 | 5.91 | 0.2 | -16.97 |
| SCB | 0.53 | 21.73 | 0.67 | -0.09 | 0.16 | -7.28 | 0.3 | 31.82 | 0.28 | 13.08 | 0.29 | -11.57 |
| PRD | 0.36 | 16.21 | 0.44 | -0.46 | 0.22 | -2.29 | 0.17 | 39.89 | 0.18 | 12 | 0.15 | -1.18 |
| China | 0.60 | 10.03 | 0.5 | -0.36 | 0.27 | -3.68 | 0.36 | 30.54 | 0.29 | 0.64 | 0.27 | -29.44 |

(b)

| | $O_3$ | | CO | | $NO_2$ | | $SO_2$ | | $PM_{2.5}$ | | $PM_{10}$ | |
|---|---|---|---|---|---|---|---|---|---|---|---|---|
| | R | MB | R | MB | R | MB | R | MB | R | MB | R | MB |
| JJJ | 0.82 | -12.54 | 0.62 | 0.21 | 0.32 | 17.85 | 0.51 | 128.82 | 0.48 | 11.98 | 0.38 | -33.71 |
| YRD | 0.52 | -12.67 | 0.51 | 0.24 | 0.27 | 5.97 | 0.38 | 68.96 | 0.24 | 13.15 | 0.23 | -9.32 |
| SCB | 0.53 | 18.49 | 0.7 | 0.00 | 0.22 | -5.25 | 0.39 | 40.55 | 0.32 | 21.7 | 0.33 | -2.73 |
| PRD | 0.34 | 13.76 | 0.45 | -0.42 | 0.26 | -0.29 | 0.19 | 47.15 | 0.21 | 16.25 | 0.18 | 3.52 |
| China | 0.61 | 5.63 | 0.51 | -0.27 | 0.28 | -0.61 | 0.38 | 41.34 | 0.31 | 6.83 | 0.29 | -24.35 |

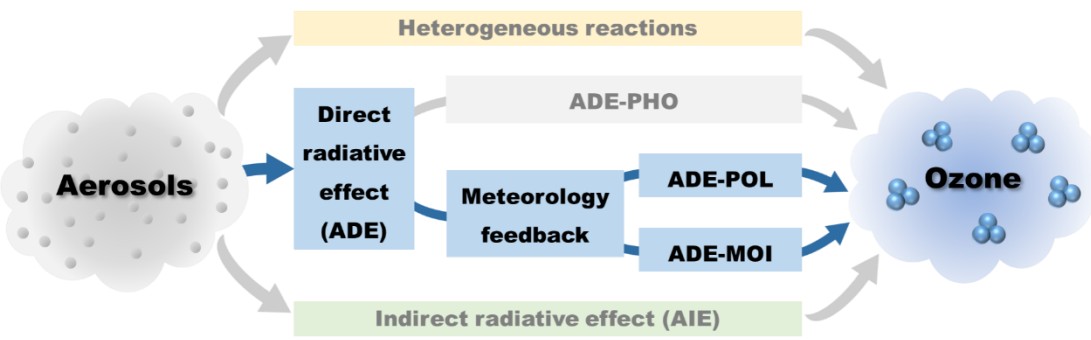

**Figure 1: The mechanism of aerosols affecting ozone. The main topic of this paper has been marked as blue lines and blocks.**

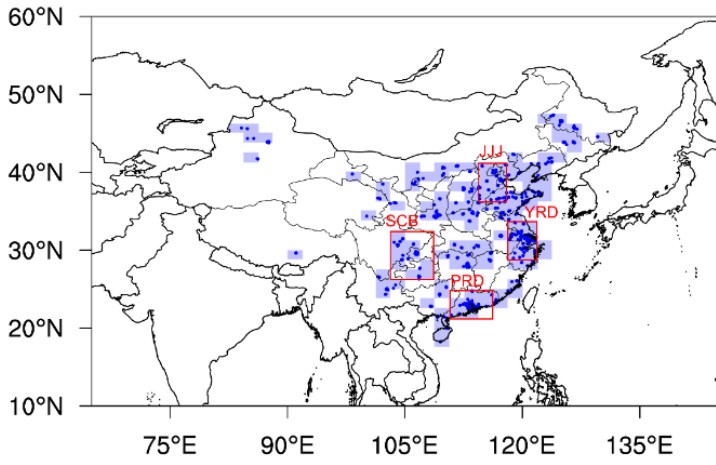

**Figure 2: Monitoring sites (blue dots), model grid-cells including the monitoring sites (pale blue squares) and the location of the four selected regions for further analysis (red grids): Jing-Jin-Ji (JJJ), Yangzi River Delta (YRD), Pearl River Delta (PRD) and Sichuan Basin (SCB).**

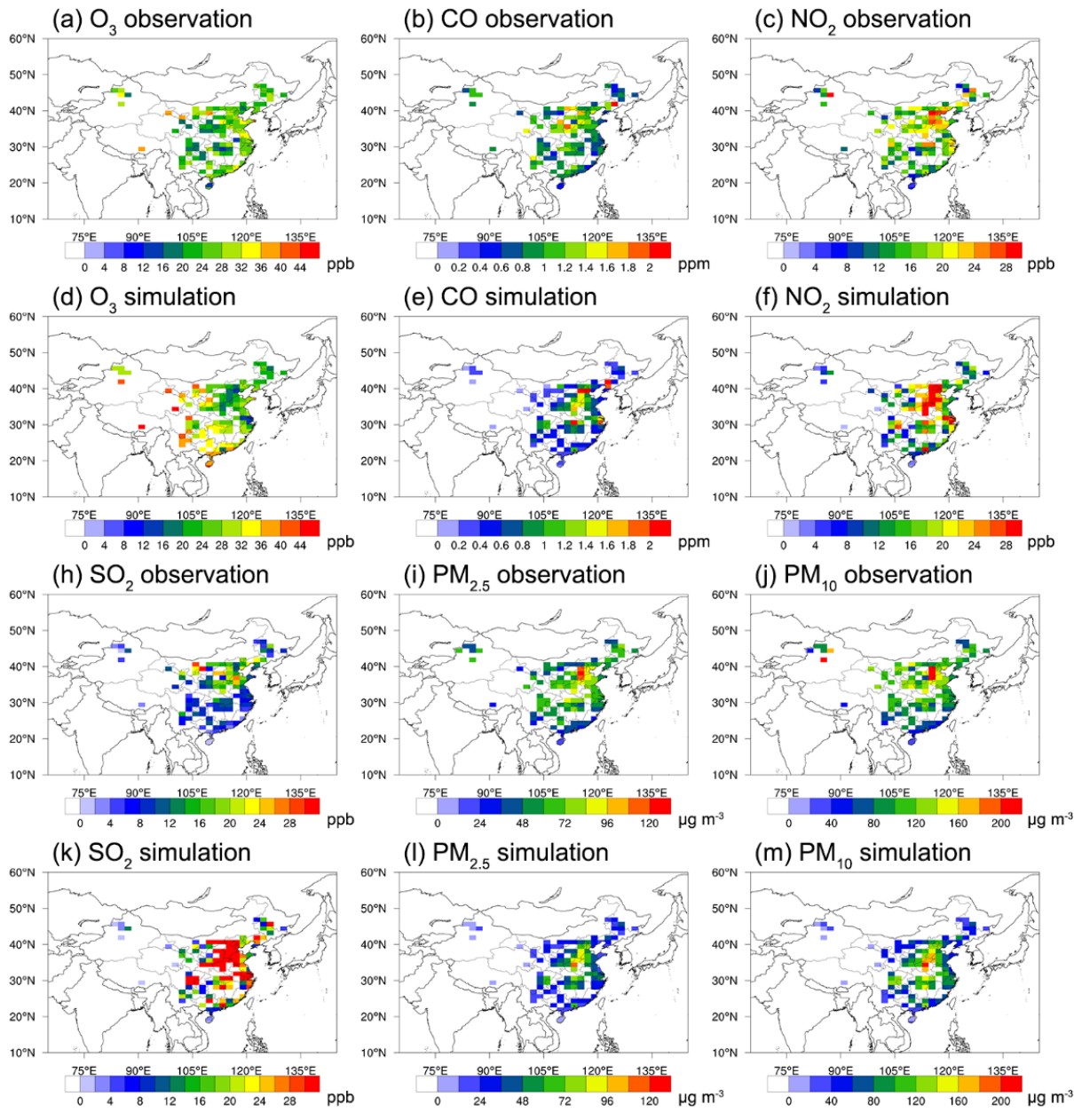

Figure 3: The observed and simulated (EXP$_{radon}$) annual average concentrations of (a, d) O$_3$, (b, e) CO, (c, f) NO$_2$, (h, k) SO$_2$, (i, l) PM$_{2.5}$ and (j, m) PM$_{10}$ in the model grid-points in 2014.


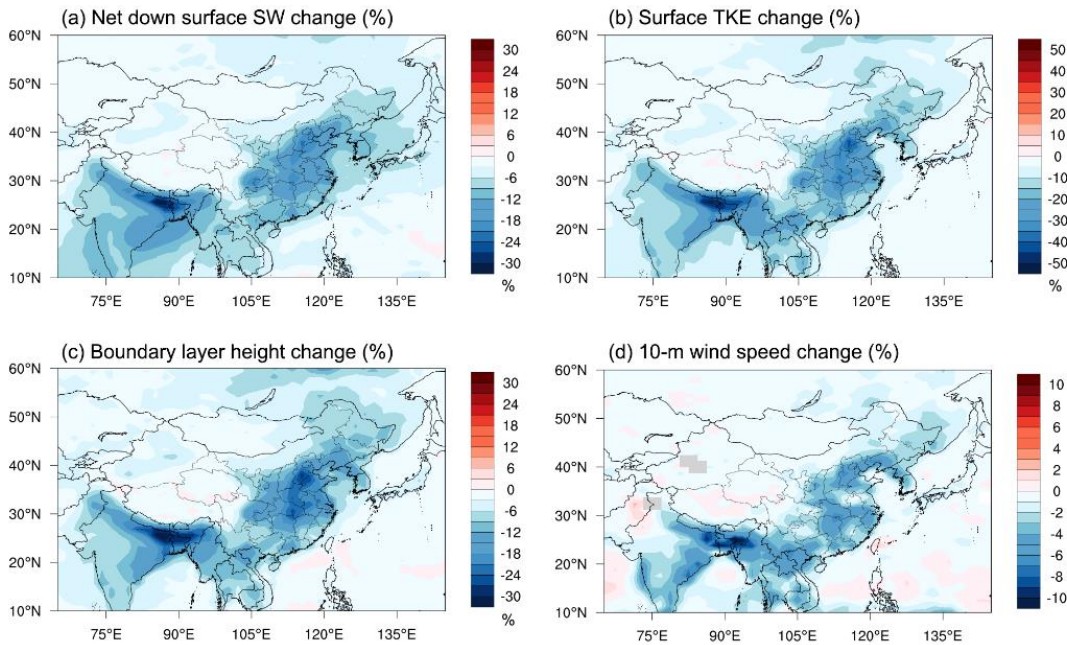

**Figure 4: Changes in (a) net down surface shortwave radiation, (b) turbulent kinetic energy (TKE), (c) boundary layer height, (d) wind due to aerosol direct radiative effect. Differences are calculated as (EXP$_{radon}$-EXP$_{radoff}$)/ EXP$_{radoff}$, averaged over one year.**

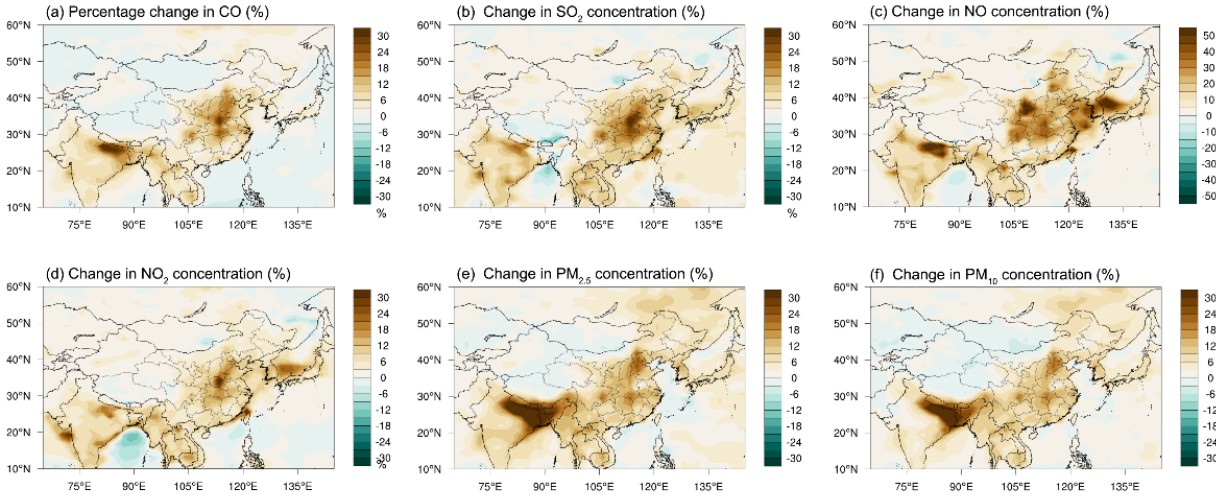

**Figure 5: Changes of (a) CO, (b) SO₂, (c) NO and (d) NO₂, (e) PM2.5 and (f) PM10 concentration due to aerosol direct radiative effect. Differences are calculated as (EXP$_{radon}$-EXP$_{radoff}$)/ EXP$_{radoff}$, averaged over one year.**


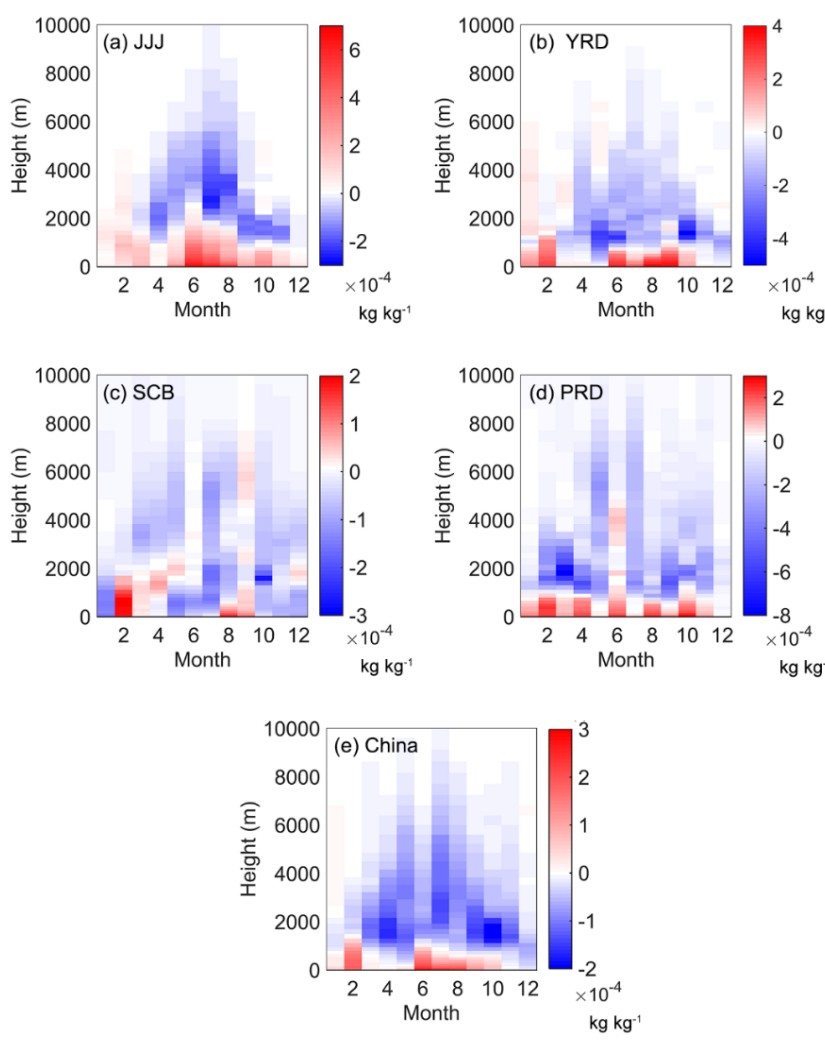

**Figure 6: Monthly changes of specific humidity in (a) Jing-Jin-Ji, (b) Yangtze river delta, (c) Sichuan Basin, (d) Pearl river delta, and (e) China due to aerosol direct radiative effect. Differences are calculated as the monthly mean of EXP$_{radon}$ minus EXP$_{radoff}$.**


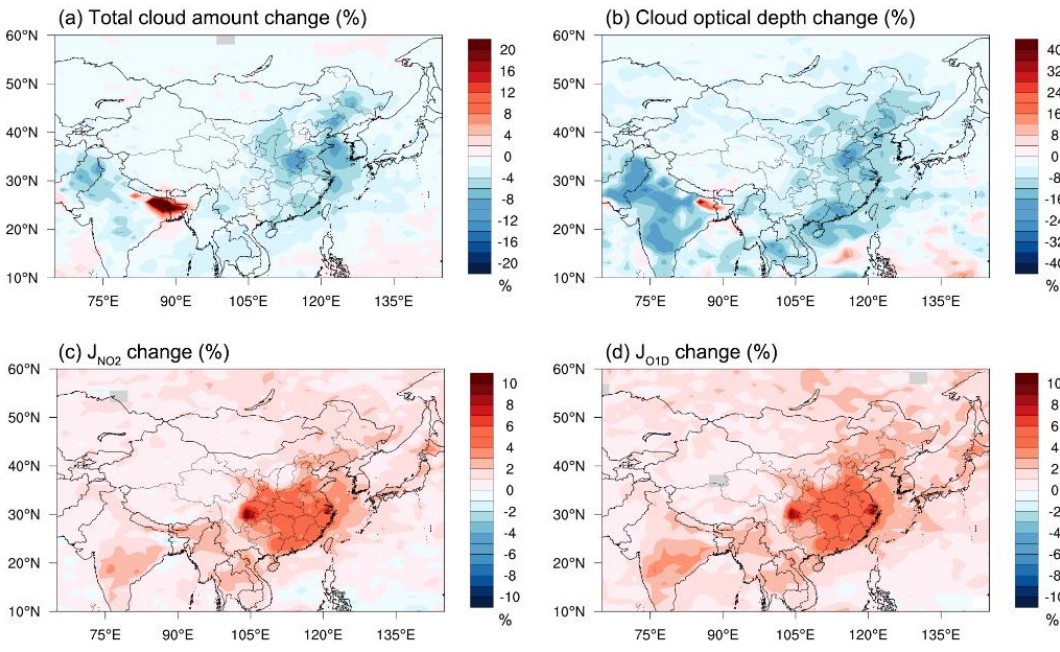

**Figure 7: Changes of (a) total cloud amount, (b) cloud optical depth, (c) J$_{NO2}$, and (d) J$_{O1D}$ due to aerosol direct radiative effect. Differences are calculated as (EXP$_{radon}$-EXP$_{radoff}$)/ EXP$_{radoff}$, averaged over one year.**

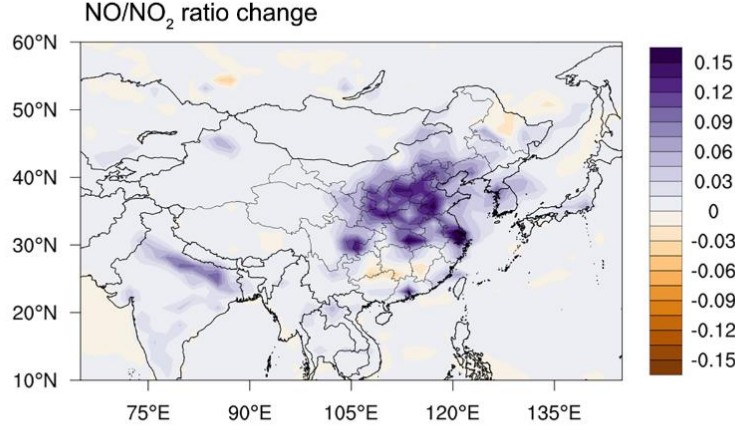


**Figure 8: Annual average change of NO/NO₂ due to aerosol direct radiative effect. Differences are calculated as the annual mean of EXP_radon minus EXP_radoff.**

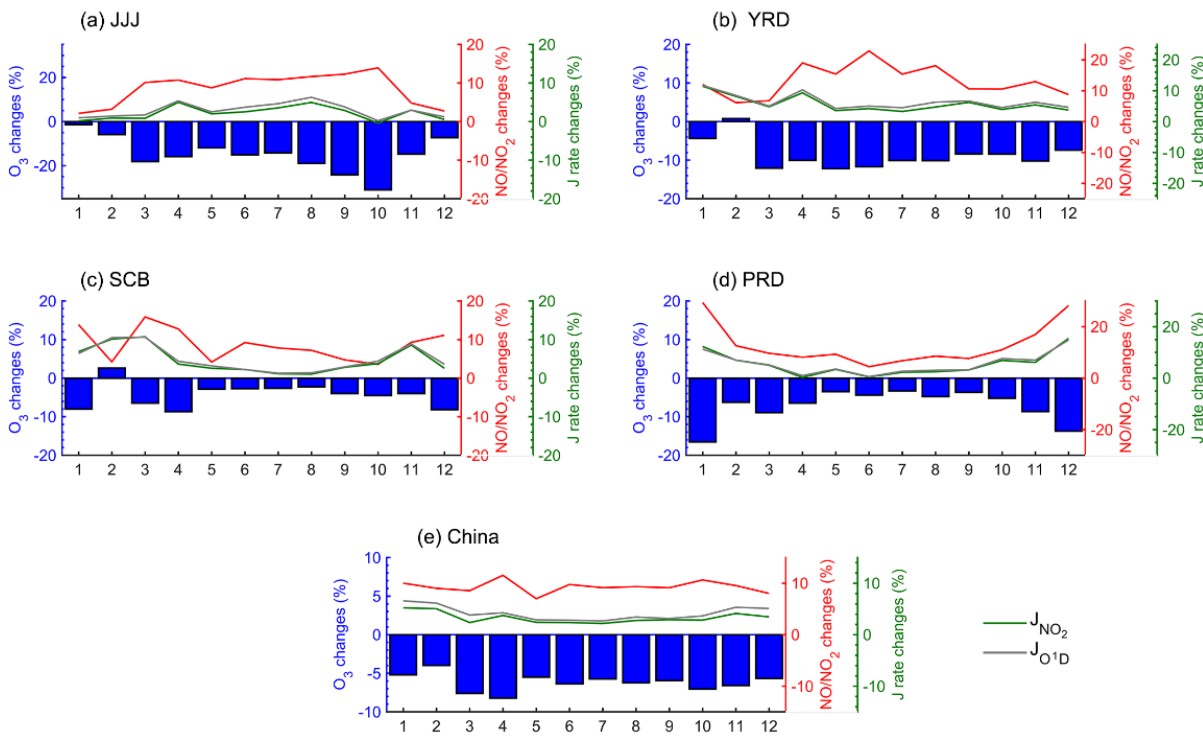

**Figure 9: Monthly variation of O₃ concentration changes, NO/NO₂ ratio changes, J_{NO2} and J_{O1D} changes in (a) Jing-Jin-Ji, (b) Yangtze river delta, (c) Sichuan Basin, (d) Pearl river delta, and (e) China due to aerosol direct radiative effect. Differences are calculated as (EXP_{radon}-EXP_{radoff})/ EXP_{radoff}, averaged over one year.**

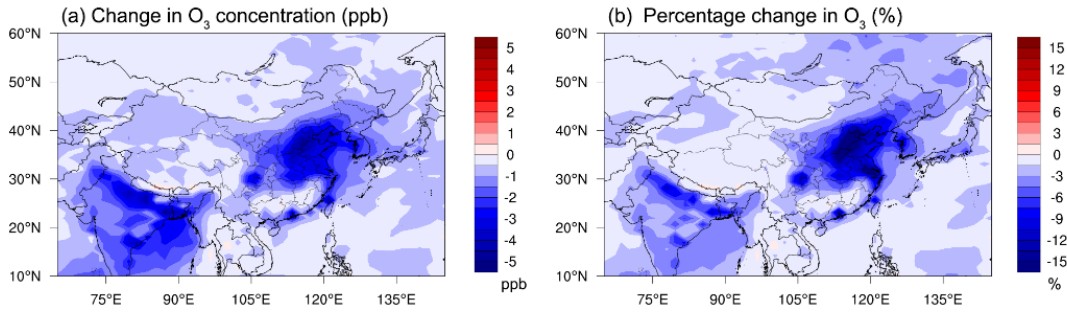

**Figure 10: Annual average change of ozone due to aerosol direct radiative effect (a) absolute changes (ppb) are calculated as EXP$_{radon}$ minus EXP$_{radoff}$, (b) percentage changes (%) are calculated as (EXP$_{radon}$-EXP$_{radoff}$)/ EXP$_{radoff}$.**