# Peer review of "A study of the effect of aerosols on surface ozone through meteorology feedbacks over China"

_Atmospheric Chemistry and Physics, 2020_

## Referee Comment (RC1) · Anonymous Referee #2 · 10 Nov 2020

The author has responded to the reviewer's request and revised the text. It is recommended that the editor receive this article.

---

## Author Comment (AC1) · 9 Dec 2020

We thank the reviewer for the approval. We would also like to thank him/her for the previous valuable comments, which have greatly improved our manuscript.

---

## Referee Comment (RC2) · Anonymous Referee #3 · 31 Jan 2021

The current paper studies the effect of aerosol feedback on O3 concentration in China specifically through ADE-POL and ADE-MOI mechanisms. The paper is well written and results have been adequately discussed. I have some comments which if included will further bolster the manuscript.

1. The increase in RH in PBL due to reduced vertical transport of water vapor results in reduced cloud cover and hence resulting in higher short-wave radiation reaching ground surface. On the contrary increased RH in PBL will result in higher aerosol hygroscopic growth thereby resulting in further increased aerosol concentration which will result in higher extinction of short-wave radiation. Can the authors address this in the manuscript?

2. The authors mention that the surface pollutant concentrations increase due to reduced shortwave radiation, however the authors exclude any discussions on VOC concentrations. Do the VOC concentrations too increase due to reduced shortwave radiation? In that case both NOx and VOC concentration being the major precursors for O3 increase, what explains the decrease in O3 concentrations?

3. The authors are suggested to either use the words reduced and increased or use symbols – and +. Refrain from using both.

---

## Author Comment (AC2) · 26 Feb 2021

The current paper studies the effect of aerosol feedback on $O_3$ concentration in China specifically through ADE-POL and ADE-MOI mechanisms. The paper is well written and results have been adequately discussed. I have some comments which if included will further bolster the manuscript.

*Response:* We thank referee #3 for careful reading and useful comments. We have responded to each specific comment in blue below. Please note that line numbers given below refer to the marked-up version of the manuscript.

1. The increase in RH in PBL due to reduced vertical transport of water vapor results in reduced cloud cover and hence resulting in higher short-wave radiation reaching ground surface. On the contrary increased RH in PBL will result in higher aerosol hygroscopic growth thereby resulting in further increased aerosol concentration which will result in higher extinction of short-wave radiation. Can the authors address this in the manuscript?

*Response:* We agree that it would be more accurate to address the effect on aerosol hygroscopic growth and the overall effect on shortwave radiation in the manuscript. We have modified line 208-219 as follows:

"When more water vapor was trapped in the lower troposphere, there would be less moisture to form cloud in the upper layers (Allen et al., 2019). The annual average cloud amount decreases by 4% due to aerosol effects on radiation over the whole country (Fig. 7). The area with the largest decline is YRD, and the percentage change is -5%. The cloud optical depth also drops by 7%-15.6% in China, with the regional distribution of changes being similar to the cloud amount changes. Clouds attenuate solar radiation, leading to diminished photolysis rates beneath the cloud (Tang et al., 2003; Voulgarakis et al., 2009a, 2009b, 2010). Therefore, the increased water vapor in PBL results in higher photolysis rates by reducing clouds. However, the increased water vapor in PBL will also enhance extinction by aerosol hygroscopic growth, which results in lower photolysis rates. Figure 7 shows that surface photolysis rates $J_{NO2}$ and $J_{O1D}$ both increase, which means, comparing to the aerosol hygroscopic growth, the aforementioned cloud reductions is the dominant effect. The national average $J_{NO2}$ and $J_{O1D}$ rose by 4.1% and

3.3%, respectively. SCB is the region with the largest increase in $J_{NO2}$ and $J_{O1D}$, with percentage increases of 8% and 7.9%, respectively. The increase in $J_{NO2}$ and $J_{O1D}$ could lead to an increase/decrease in ozone concentration."

2. The authors mention that the surface pollutant concentrations increase due to reduced shortwave radiation, however the authors exclude any discussions on VOC concentrations. Do the VOC concentrations too increase due to reduced shortwave radiation? In that case both NOx and VOC concentration being the major precursors for O3 increase, what explains the decrease in O3 concentrations?

***Response:*** Due to meteorological feedback, the annual concentration of VOC in East China has increased by about 10% (Figure R1), which is not as significant as $NO_x$ (Figure 5). Although both $NO_x$ and VOC are precursors of ozone, they are not linearly related to ozone concentration. Previous studies have found that the reduction of $NO_x$ will lead to the increase of ozone in the VOC-sensitive region, even after accounting for VOC reductions (Jhun et al., 2015; Li et al., 2018; Li et al., 2021). Wang et al. (2019) found that only when the reduction ratio of VOCs/$NO_x$ is greater than 2:1, can ozone be effectively reduced in the VOC-sensitive region of China. It can be inferred that when $NO_x$ and VOC increase in the VOC-sensitive region, especially when $NO_x$ (NO) increase dominates, ozone concentration may decrease.

**References**

Jhun, I., Coull, B.A., Zanobetti, A., Koutrakis, P., 2015. The impact of nitrogen oxides concentration decreases on ozone trends in the USA. Air Qual. Atmos. Heal. 8, 283–292. https://doi.org/10.1007/s11869-014-0279-2

Li, K., Jacob, D.J., Liao, H., Shen, L., Zhang, Q., Bates, K.H., 2018. Anthropogenic drivers of 2013–2017 trends in summer surface ozone in China. Proc. Natl. Acad. Sci. 201812168. https://doi.org/10.1073/pnas.1812168116

Li, M., Wang, T., Shu, L., Qu, Y., Xie, M., Liu, J., Wu, H., Kalsoom, U., 2021. Rising surface ozone in China from 2013 to 2017: A response to the recent atmospheric warming or pollutant controls? Atmos. Environ. 246, 118130. https://doi.org/10.1016/j.atmosenv.2020.118130

Wang, N., Lyu, X., Deng, X., Huang, X., Jiang, F., Ding, A., 2019. Aggravating O₃ pollution due to $NO_x$ emission control in eastern China. Sci. Total Environ. 677, 732–744.

https://doi.org/10.1016/j.scitotenv.2019.04.388

[Figure]

Figure R1: Changes of VOC concentration (%) due to aerosol direct radiative effect. Differences are calculated as (EXP$_{radon}$-EXP$_{radoff}$)/ EXP$_{radoff}$, averaged over one year.

3. The authors are suggested to either use the words reduced and increased or use symbols – and +. Refrain from using both.

***Response:*** Thanks for your suggestion. We use "reduced" and "increased" instead of symbols – and + in the revised manuscript.